# The liquid structure of elastin

Sarah Rauscher[1,2†], Régis Pomès[1,2]*

[1]Molecular Medicine, The Hospital for Sick Children, Toronto, Canada; [2]Department of Biochemistry, University of Toronto, Toronto, Canada

**Abstract** The protein elastin imparts extensibility, elastic recoil, and resilience to tissues including arterial walls, skin, lung alveoli, and the uterus. Elastin and elastin-like peptides are hydrophobic, disordered, and undergo liquid-liquid phase separation upon self-assembly. Despite extensive study, the structure of elastin remains controversial. We use molecular dynamics simulations on a massive scale to elucidate the structural ensemble of aggregated elastin-like peptides. Consistent with the entropic nature of elastic recoil, the aggregated state is stabilized by the hydrophobic effect. However, self-assembly does not entail formation of a hydrophobic core. The polypeptide backbone forms transient, sparse hydrogen-bonded turns and remains significantly hydrated even as self-assembly triples the extent of non-polar side chain contacts. Individual chains in the assembly approach a maximally-disordered, melt-like state which may be called the liquid state of proteins. These findings resolve long-standing controversies regarding elastin structure and function and afford insight into the phase separation of disordered proteins.
DOI: https://doi.org/10.7554/eLife.26526.001

*For correspondence:
pomes@sickkids.ca

Present address: †Department of Chemical and Physical Sciences, University of Toronto Mississauga, Mississauga, Canada

## Introduction

The elasticity of skin, lungs, major arteries, and other vertebrate tissues is imparted by the fibrous structural protein, elastin. Networks of elastic fibres are formed in the extracellular matrix from the monomeric precursor, tropoelastin. Elastic fibres are subject to minimal turnover during a lifetime and are highly durable under repetitive physiological strain (*Shapiro et al., 1991*; *Davis, 1993*). For example, elastic fibres in the arterial wall enable the tissue to undergo over two billion cycles of extension and relaxation to smooth the flow of blood down the arterial tree, largely without mechanical failure. The remarkable durability and functional resilience of elastic fibres arise from intrinsic features of the monomer. Tropoelastin is a 60 kDa modular protein composed of alternating hydrophobic and cross-linking domains (*Muiznieks et al., 2010*). Although both types of domains contribute to the proper supramolecular assembly and the mechanical properties of the polymeric elastin network, the cross-linking domains are understood to bestow cohesiveness and durability to the material, whereas the hydrophobic domains confer the propensities for self-assembly and elastic recoil (*Rauscher and Pomès, 2012*). Solutions of tropoelastin and elastin-like peptides (ELPs) self-aggregate via liquid-liquid phase separation upon increasing temperature, a process known as coacervation. This process has been shown to be primarily driven by the self-association of hydrophobic domains (*Muiznieks et al., 2010*; *Urry et al., 1974*; *Bellingham et al., 2001*; *Toonkool et al., 2001*). The capacity for temperature-controlled self-assembly make elastin-like peptides well-suited for biomaterials applications (*Almine et al., 2010*) and drug delivery (*Shi et al., 2013*).

Despite the biological importance of elastin and eighty years of study using a myriad of biophysical techniques (*Meyer and Ferri, 1937*), neither the molecular basis of self-assembly nor the structure of the self-assembled state are known. Numerous structural models of elastin have been proposed, which span a range from highly-ordered (*Venkatachalam and Urry, 1981*) to maximally-disordered (*Hoeve and Flory, 1958*; *Flory, 1974*) and emphasize either the hydrophobic effect (*Venkatachalam and Urry, 1981*; *Weis-Fogh and Anderson, 1970*; *Gray et al., 1973*; *Li et al.,*

2001a) or conformational entropy (*Hoeve and Flory, 1958*; *Flory, 1974*; *Dorrington et al., 1975*) as the dominant contribution to the elastic recoil force.

High conformational entropy is the key feature of the earliest model proposed for elastin's structure: the random network model (*Hoeve and Flory, 1958*; *Flory, 1974*). Based on thermoelasticity measurements indicating that elastin's recoil force is almost entirely entropic, Hoeve and Flory postulated that elastin's structure is an isotropic, rubber-like polymer network consisting of cross-linked, random chains (*Hoeve and Flory, 1958*; *Flory, 1974*). However, thermoelasticity measurements on elastin samples were carried out using alcohol diluents (*Hoeve and Flory, 1958*; *Flory, 1974*; *Dorrington and McCrum, 1977*; *Andrady and Mark, 1980*) and therefore rely on the assumption that alcohols do not significantly perturb the structure of elastin. The validity of this assumption has been questioned (*Weis-Fogh and Anderson, 1970*; *Ellis and Packer, 1976*; *Gosline, 1978*; *Chalmers et al., 1999*) because the sequence composition of elastin is unusually enriched in hydrophobic residues, which likely interact with alcohols. Other types of mechanical studies have been carried out in a wide variety of solvents (*Weis-Fogh and Anderson, 1970*; *Lillie and Gosline, 2002*; *Silverstein et al., 2015*). Whether or not the idealized random network model applies to the functional state of elastin is not known and remains controversial.

In contrast to the random network model, specific secondary structure preferences and significant burial of non-polar groups are common features of several other structural models of elastin (*Venkatachalam and Urry, 1981*; *Weis-Fogh and Anderson, 1970*; *Gray et al., 1973*; *Li et al., 2001a*; *Gosline, 1978*). Elastin's highly hydrophobic sequence led several groups to propose that the hydrophobic effect, rather than conformational entropy, is the major driving force of elastic recoil (*Weis-Fogh and Anderson, 1970*; *Gray et al., 1973*; *Li et al., 2001a*; *Gosline, 1978*). Several models were proposed in which non-polar side chains are arranged to exclude water molecules; the most ordered of these models is the β-spiral, which consists of repeated β-turns (*Venkatachalam and Urry, 1981*). While the β-spiral model has been shown to be unstable (*Li et al., 2001a*) spectroscopic data are consistent with the presence of β-turns (*Muiznieks et al., 2010*; *Tamburro et al., 2003*).

To go beyond these largely qualitative and seemingly contradictory models, high-resolution structural information is required. The conformational heterogeneity and self-association of elastin have impeded crystallographic and spectroscopic investigations and present a significant sampling challenge to molecular simulations (*Rauscher and Pomès, 2010a*; *Rauscher et al., 2009*). Accordingly, most previous computational studies of elastin-like peptides have been limited to molecular dynamics (MD) simulations of peptide monomers, starting with simulations of ~100 ps in vacuo (*Chang and Urry, 1988*; *Wasserman and Salemme, 1990*; *Lelj et al., 1992*) and moving on to simulations of ~10–100 ns in explicit water (*Li et al., 2001a*; *Silverstein et al., 2015*; *Rauscher et al., 2009*; *Li et al., 2001b*; *Schreiner et al., 2004*; *Rauscher et al., 2006*; *Krukau et al., 2007*; *Glaves et al., 2008*; *Li et al., 2014*; *Condon et al., 2017*; *Reppert et al., 2016*; *Tang et al., 2016*). Although some of these studies examined assemblies comprising between 2 and 6 peptides (*Rauscher et al., 2006*; *Li et al., 2014*; *Condon et al., 2017*), to our knowledge simulations of larger aggregates of elastin-like (or, indeed, other intrinsically-disordered) peptides have never been reported. Moreover, we have shown that attaining statistically-converged sampling of disordered elastin-like peptides necessitates simulation times in the microsecond time-range (*Rauscher et al., 2009*; *Rauscher and Pomès, 2010b*).

To examine the structural and physico-chemical basis for the self-assembly of elastin, we use MD simulations on a massive scale. For the sake of computational feasibility, we neglect cross-linking domains and consider a repetitive sequence, (GVPGV)$_7$, modeled on hydrophobic domains of elastin. Although cross-linking domains are required to form elastomeric materials, they do not undergo coacervation on their own and are not required for coacervation (*Urry et al., 1974*; *Bellingham et al., 2001*). Peptides based on GVPGV and other repetitive motifs from the hydrophobic domains of tropoelastin have been shown to coacervate in the absence of cross-linking domains (*Urry et al., 1974*; *Miao et al., 2003*; *Muiznieks et al., 2014*). As such, our model system is not designed to capture all the properties of elastin. Instead, we focus on the phase separation of hydrophobic elastin-like domains, which also provides insight into the structural basis of elasticity. All-atom MD simulations of a monomer and an aggregate of 27 peptides, with a combined sampling time exceeding 200 μs, are used to provide the first atomistic description of the conformational ensemble of an elastin-like peptide successively in solution and in aggregated form.

## Results and discussion

### Peptide chain dimensions before and after self-assembly

We first compare the ensembles of the elastin-like peptide in solution (single chain, SC) and as an aggregate (multi-chain, MC) with respect to chain dimensions (*Figure 1*). Both in solution and in the aggregate, the peptide chains sample heterogeneous, disordered structural ensembles, without a unique, preferred conformation. The ensembles differ significantly with respect to chain dimensions, with aggregated chains being much more expanded on average than the single chain in solution. Not only is the average radius of gyration, $R_g$, higher for aggregated chains, the variance of $R_g$ is also larger, indicative of a more heterogeneous underlying conformational ensemble.

High conformational disorder in the aggregate is corroborated by multiple, independent experimental observations on elastin: elastin fibres are optically isotropic (*Aaron and Gosline, 1980*); the backbone carbonyl order parameter is less than 0.1 (*Pometun et al., 2004*); carbon chemical shifts are consistent with random coil secondary structure (*Pometun et al., 2004*); and neutron scattering experiments show that the polypeptide chains in elastin are highly mobile (*Perticaroli et al., 2015*). High conformational entropy underpins the random network model of elastin structure and function

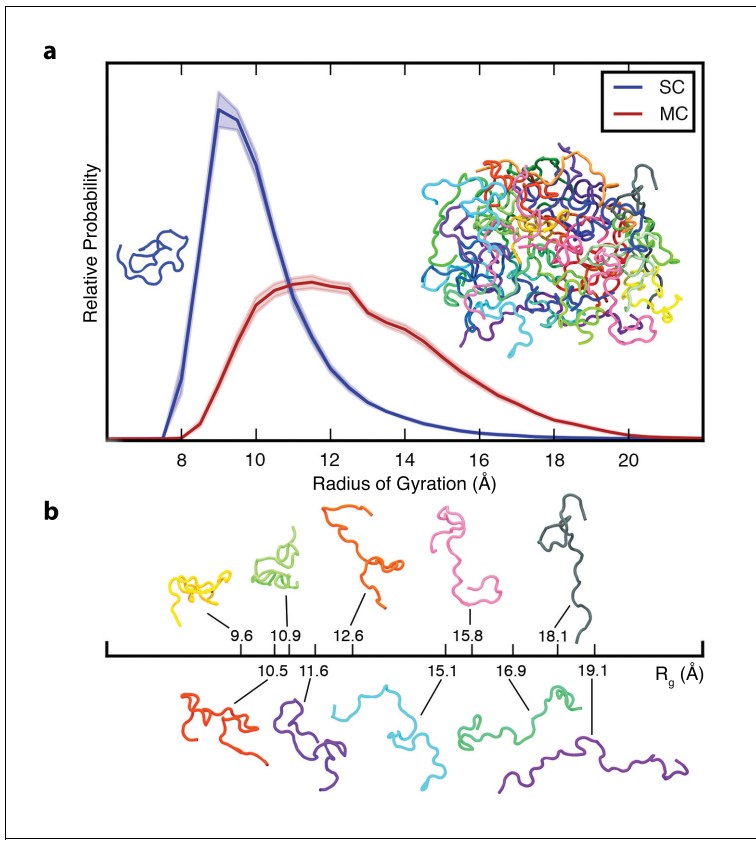

**Figure 1.** Ensemble-averaged polypeptide chain dimensions. (**a**) Probability distribution of the radius of gyration, $R_g$, for SC (blue) and MC (red). Insets show representative backbone conformations of peptide monomer (left) and aggregate (right). (**b**) Aggregated chains are colored individually and ten of them are also shown below with their corresponding $R_g$ to illustrate the conformational heterogeneity. In all figures, error bars indicate standard error. All the results reported in this and subsequent figures were obtained at 298 K unless otherwise noted.

DOI: https://doi.org/10.7554/eLife.26526.002

The following figure supplements are available for figure 1:

**Figure supplement 1.** Formation and collapse of the aggregate.
DOI: https://doi.org/10.7554/eLife.26526.003

**Figure supplement 2.** Ensemble-averaged polypeptide chain dimensions with the TIP4P-D water model.
DOI: https://doi.org/10.7554/eLife.26526.004

initially proposed by Hoeve and Flory (*Hoeve and Flory, 1958*; *Flory, 1974*). In this model, conformational entropy decreases when the chains are stretched and increases upon relaxation, thereby driving elastic recoil. Consistent with the random network model, we find that the peptide chains in the aggregate are highly disordered.

## Disordered, but not random: a probabilistic description of intra- and interchain interactions

In order to characterize the complex conformational landscape of the disordered ELP chains, we obtained a statistical picture of the different conformational states and interactions accessible to the chains in solution and in the aggregate. Statistical maps of the two types of peptide-peptide interactions, backbone hydrogen bonds and non-polar side chain contacts, reflect highly-disordered conformational ensembles (*Figure 2*). Secondary structure is sparse and limited to transient (sub-ns) hydrogen-bonded turns between residues close in sequence (near-diagonal elements in *Figure 2a, b*). The most populated structures are VPGV and GVGV β-turns. The propensity of each of these local interactions, which peaks at 20%, is remarkably well conserved upon aggregation (*Table 1*), while the total number of peptide-peptide hydrogen bonds per chain increases only moderately (*Figure 2c*). This evidence for fluctuating β-turns is consistent with spectroscopic data (*Muiznieks et al., 2010*; *Tamburro et al., 2003*) but not with the β-spiral, a highly-ordered structural model of elastin which requires that repeated β-turns be formed simultaneously, (*Venkatachalam and Urry, 1981*) which is never observed in the simulations. As such, our results reconcile spectroscopic evidence for local secondary structure (*Muiznieks et al., 2010*; *Tamburro et al., 2003*; *Ohgo et al., 2012*) with global conformational disorder.

Even as it preserves local structural propensities, self-aggregation results in the replacement of non-local intramolecular interactions by intermolecular interactions (*Figure 2*). In particular, the non-local non-polar contacts that characterize the collapsed isolated chain (*Figure 2d*) give way to non-specific interactions with neighboring peptides (*Figure 2e*). In this process, the bulky valine side chains contribute over 70% to all non-polar contacts in the monomeric and assembled states (*Figure 2—figure supplement 1*). The average number of non-polar contacts per chain nearly triples upon self-assembly (*Figure 2f*) with a commensurate decrease in the hydration of non-polar side-chains (*Figure 3b*), indicating that the hydrophobic effect strongly contributes to the formation and the structure of the aggregate. Accordingly, the hydrophobic effect is the major driving force for elastic recoil in several earlier models of elastin (*Venkatachalam and Urry, 1981*; *Weis-Fogh and Anderson, 1970*; *Gray et al., 1973*; *Li et al., 2001a*; *Gosline, 1978*). Contrary to these models, however, significant hydrophobic burial is achieved even in the absence of a well-ordered structure.

## Hydration and disorder of the polypeptide backbone

While self-assembly effectively buries non-polar side chains (*Figure 2e,f*), disorder of the polypeptide backbone precludes the formation of a water-excluding hydrophobic core. Since there is only a moderate amount of secondary structure, a majority of backbone peptide groups do not form peptide-peptide hydrogen bonds (*Figure 2c*). Instead, water molecules remain within the aggregate (*Figure 3a,d,e*, *Figure 4*) in order to satisfy the hydrogen bonding requirements of backbone groups. As a result, there is little loss of backbone hydration upon peptide self-assembly (*Figure 3c*), even as the side chains become dehydrated (*Figure 3b*). The high degree of hydration of the aggregate (40.0 ± 0.2% water content by mass, based on the number of water molecules hydrogen-bonded to the peptide) is consistent with experimental measurements of water content between 40% and 60% for elastin derived from various species (*Chalmers et al., 1999*). In fact, the water content of both the monomer and the aggregate is so high that the probability of any five-residue segment to be completely dehydrated is essentially zero (*Figure 3—figure supplement 2b*). These results clearly indicate that both systems lack a water-excluding hydrophobic core, a consequence of the high degree of conformational disorder and lack of extended secondary structure imparted by the high proline and glycine content (*Rauscher et al., 2006*).

The analysis of the average density of peptide and water from the center of mass of the aqueous monomer and of the peptide aggregate quantifies the presence of water throughout the system, even near the center of the peptide, where it represents over 0.2 g/cm$^3$ in both systems (*Figure 4*). The fact that this value is nearly identical in the monomer and in the aggregate suggests that internal

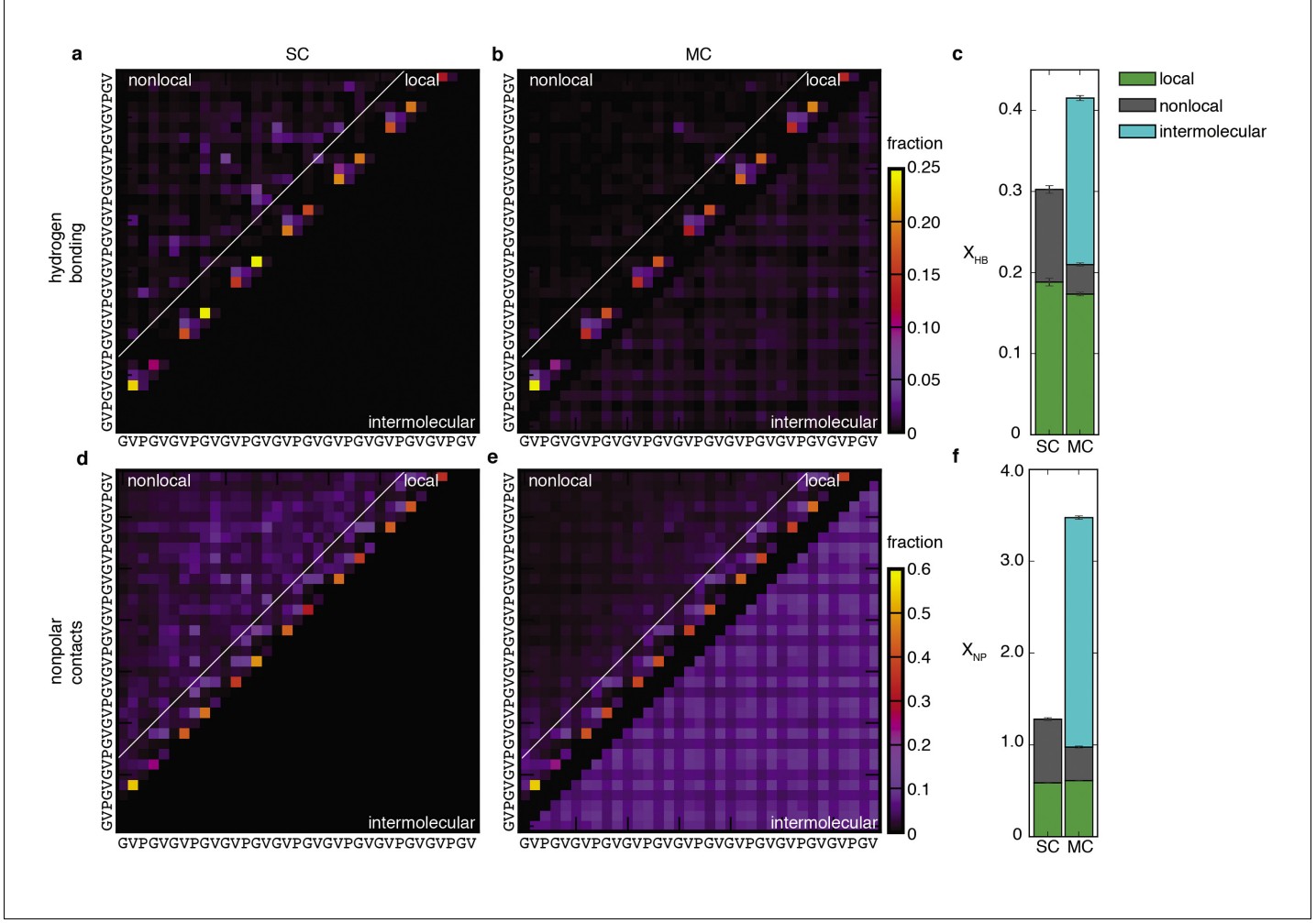

**Figure 2.** Peptide-peptide interactions. Probabilistic description of hydrogen bonding (top row) and non-polar (bottom row) interactions of SC (**a** and **d**) and MC (**b** and **e**) systems. Panels (**a**), (**b**), (**d**) and (**e**) are contact maps for pairwise interactions between residues. The color of each square indicates the fraction of conformations in the ensemble for which that interaction is present. Nearest- and next-nearest-neighbour contacts are excluded for clarity in (**d**) and (**e**). Local interactions consist of sparse backbone hydrogen bonds and corresponding non-polar contacts. Non-local interactions consist primarily of non-specific non-polar contacts between side chains. With the absence of preferred non-local interactions, the statistical picture of the conformational ensembles is remarkably simple. Upon aggregation, local structure propensities are retained as non-local hydrophobic contacts become intermolecular contacts (below the diagonal in **e**). (**c**) Average number of chain-chain hydrogen bonds per residue, $X_{HB}$. (**f**) Average number of non-polar contacts per residue, $X_{NP}$. Both $X_{HB}$ and $X_{NP}$ are the sum of intramolecular (local and non-local) and intermolecular contributions. Details of the structural analysis methods are provided in Materials and methods and ***Supplementary file 1***, Table S2.

DOI: https://doi.org/10.7554/eLife.26526.005

The following figure supplements are available for figure 2:

**Figure supplement 1.** Non-polar contacts between residue pairs.

DOI: https://doi.org/10.7554/eLife.26526.006

**Figure supplement 2.** Hydrogen bonding contact maps.

DOI: https://doi.org/10.7554/eLife.26526.007

**Figure supplement 3.** Non-polar contact maps.

DOI: https://doi.org/10.7554/eLife.26526.008

**Figure supplement 4.** Peptide-peptide interactions for the SC ensemble obtained using the TIP4P-D water model.

DOI: https://doi.org/10.7554/eLife.26526.009

**Table 1.** Populations and lifetimes of hydrogen-bonded turns.

| Turn | Population (%) | | Lifetime (ns) | |
|---|---|---|---|---|
| | SC | MC | SC | MC |
| VPGV[*] | 18 ± 1 | 16 ± 2 | 0.18 ± 0.02 | 0.19 ± 0.01 |
| VPGVG | 5.9 ± 0.9 | 4.6 ± 0.7 | 0.9 ± 0.3 | 0.8 ± 0.1 |
| PGV | 1.5 ± 0.1 | 2.0 ± 0.2 | 0.008 ± 0.002 | 0.009 ± 0.001 |
| PGVG | 2.3 ± 0.2 | 2.9 ± 0.3 | 0.30 ± 0.05 | 0.39 ± 0.03 |
| GVGV | 20 ± 3 | 17 ± 2 | 0.15 ± 0.02 | 0.24 ± 0.07 |

[*]A hydrogen-bonded turn refers to a hydrogen bond between the first and last residue in the sequences shown. For example, the VPGV turn has a hydrogen bond between valine 1 and valine 4.
DOI: https://doi.org/10.7554/eLife.26526.010

hydration is independent of size and would also be observed in larger and smaller aggregates. The presence of a plateau in water density indicates that the interior of the aggregate is homogeneous and that the aggregate is large enough to lead to the emergence of bulk-like properties expected of a separate liquid phase.

## The liquid structure of elastin

The structural ensemble of the aggregate is disordered (*Figure 1*), yet contains a significant propensity for secondary structure in the form of transient hydrogen-bonded turns (*Figure 2b*, *Table 1*) and is highly hydrated (*Figures 3* and *4*). To understand why aggregation induces peptide expansion (*Figure 1a*), we examine our results in terms of solvent quality. In a poor solvent, solute-solute interactions are energetically more favorable than solvent-solute interactions, leading to a collapse of the polymer chain. Inversely, solvent-solute interactions are preferred in a good solvent, leading to chain expansion. In the ideal limit between the two (the so-called 'θ-solvent'), there is no preference for one type of interaction over the other. In this minimally-constrained state, the polymer becomes maximally disordered. The Flory theorem states that in the liquid, phase-separated state, the 'polymer melt', the polymer chains make extensive interactions with one another and become their own θ-solvent, since intramolecular and intermolecular interactions are chemically indistinguishable; as a result of which the chains reach a state of maximal disorder. (*Flory, 1953*; *Flory, 1969*; *Flory, 1949*) Although disordered protein aggregates have been hypothesized to resemble polymer melts, (*Fields et al., 1992*; *Pappu et al., 2008*) whether or not they satisfy the Flory theorem is unknown.

The aggregated ELP chains are extensively solvated by one another (*Figure 2c,f*), suggesting that they may approach the ideal, θ-solvent limit. To quantify the extent to which the chains approach this ideal limit, we compared the intrachain distance scaling of the conformational ensembles in the simulations to the scaling expected for the ideal, θ-solvent limit (*Figure 5*). As a model for this ideal state, we use the residue-specific model for a random-coil polypeptide developed by Flory and co-workers (*Flory, 1969*; *Miller et al., 1967*). In their model, rotation about the backbone dihedral angles, φ and ψ, is treated as independent of the conformation of neighboring residues in the chain. We update their method to include residue-specific transformation matrices derived directly from the simulation data (see the detailed description of the method in *Supplementary file 1*). The internal distance scaling profile for the Flory model of the ideal state is fit by a power law with exponent $\alpha = 0.54$, which is very similar to the expected dimensions of an ideal Flory random coil homopolymer, for which $\alpha = 0.5$ (*Flory, 1969*). Similarly, the dimensions of the chains in the aggregate, with an exponent $\alpha = 0.46$, closely approach the ideal limit as well. In contrast, the dimensions of the ensemble in solution, with an exponent of $\alpha = 0.28$, are consistent with those of a polymer in a poor solvent, for which $\alpha = 1/3$. This result is expected, given the highly hydrophobic composition of the ELP sequence, and the fact that water is a poor solvent for hydrophobic residues. The internal distance scaling profile of the chain in solution exhibits a slight upturn at large sequence separations. This deviation from the behavior expected for a perfectly collapsed, globular homopolymer likely arises from the fact that the conformational landscape of the ELP in solution is more complex than that of a simple homopolymer in a poor solvent due to the presence of specific, highly-populated turns (*Table 1* and *Figure 2b*).

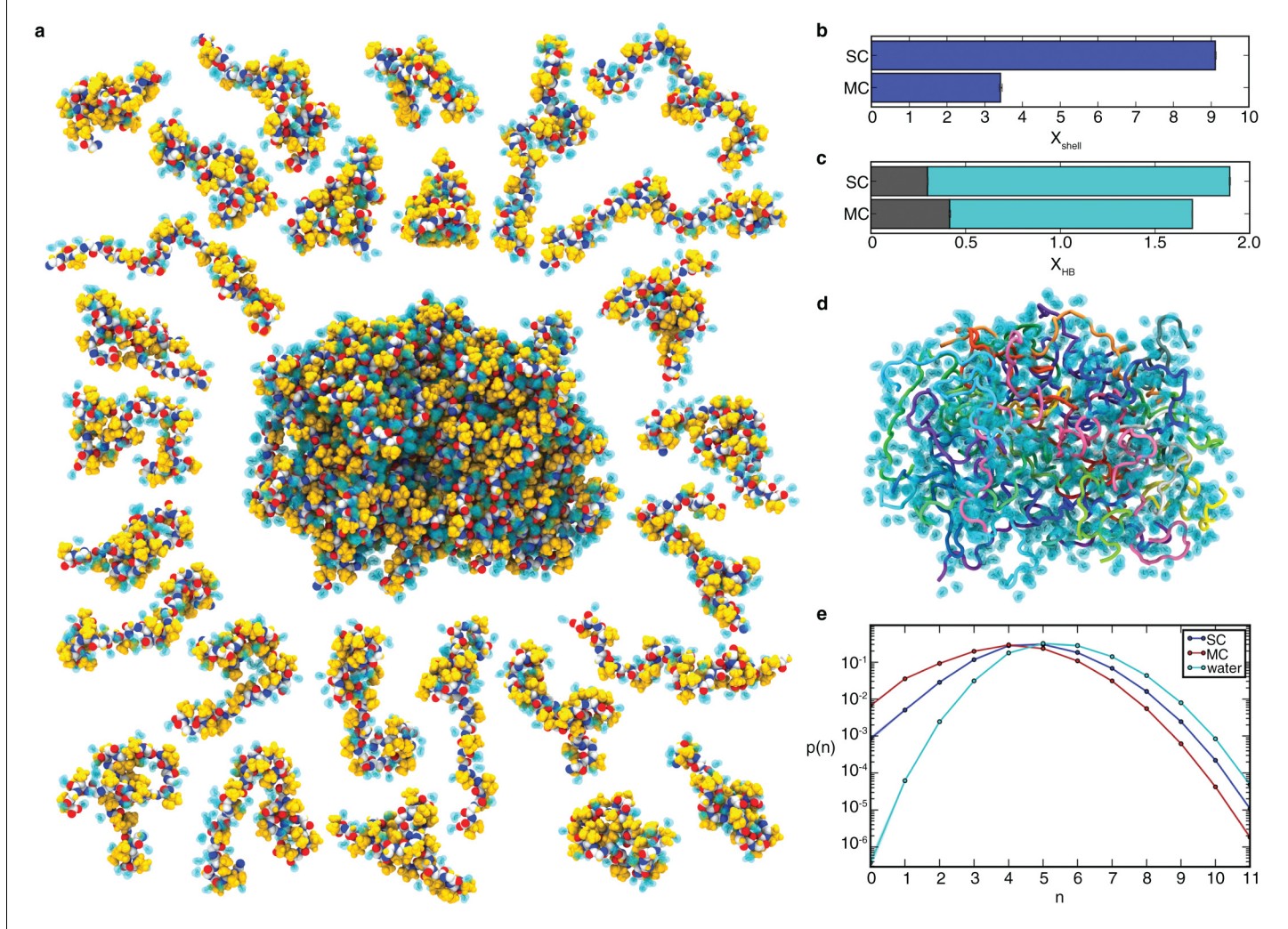

**Figure 3.** Peptide hydration in the liquid-like aggregate. (**a**). Representative conformation of the aggregate with non-polar side chains (yellow), peptide backbone (oxygen, red; carbon, white; nitrogen, blue), and hydrogen-bonded water molecules (cyan). Peptide chains are shown individually on the periphery with bound water molecules. (**b**) Average number of water molecules in the hydration shell per residue, $X_{shell}$, for SC and MC systems (see *Supplementary file 1*, Table S2). (**c**) Average number of hydrogen bonds per residue, $X_{HB}$. $X_{HB}$ is the sum of peptide-peptide hydrogen bonds (grey) and peptide-water hydrogen bonds (cyan), for both the SC and MC systems. (**d**) Same conformation as in panel a with bound water molecules shown as a transparent surface and peptides coloured individually. (**e**) Probability distribution, $p(n)$, of water coordination number, $n$, for water molecules in the hydration shell of the SC (blue) and the MC (red) systems and in bulk water (cyan) at 298 K. Peptide-bound water molecules in the aggregate have fewer neighbors. The lines are shown to guide the eye.

DOI: https://doi.org/10.7554/eLife.26526.011

The following figure supplements are available for figure 3:

**Figure supplement 1.** Conformations after 5 μs of simulation.
DOI: https://doi.org/10.7554/eLife.26526.012

**Figure supplement 2.** Hydration of blob-sized segments.
DOI: https://doi.org/10.7554/eLife.26526.013

The dimensions of the chains in the ELP aggregate approach the dimensions predicted for maximally-disordered chains (*Figure 5*), and therefore the dimensions expected in a polymer melt. Deviation from ideality reflects the finite size of the aggregate, finite chain length, the presence of local secondary structure (*Figure 2b*), and persistent hydration (*Figures 3* and *4*). These results suggest that elastin—and polypeptide chains in general—cannot make polymer melts in the idealized, solvent-excluding sense because backbone groups must form hydrogen bonds either with each other, which leads to ordering, or with water molecules, whose presence is required for disorder. Instead,

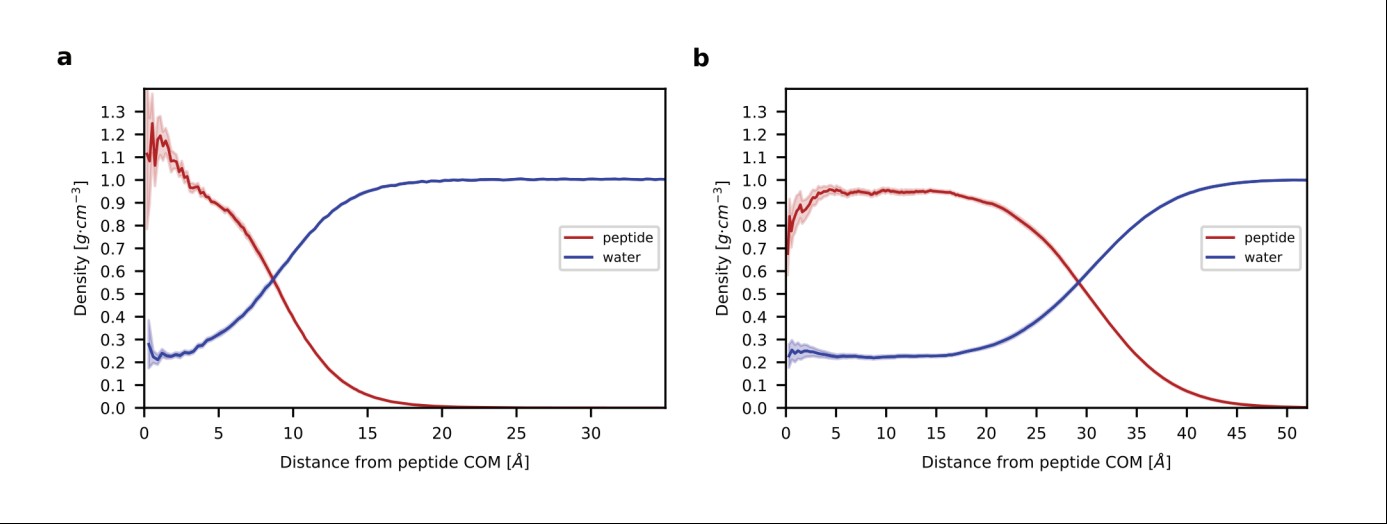

**Figure 4.** Radial density profiles. (a). Density profiles for peptide (red) and water (blue) as a function of the distance from the center of mass (COM) of the peptide is shown for the single chain system. (b) Density profiles for peptide and water as a function of the distance from the COM of the aggregate is shown for the multi-chain system. Shading in (a) and (b) indicates standard error. Note that the large width of the transition region between the homogenous interior and bulk water reflects not only the higher hydration of residues at the surface but also the asphericity of the aggregate (see *Figure 3—figure supplement 1*).

DOI: https://doi.org/10.7554/eLife.26526.014

the chains adopt a disordered state with significant backbone hydration, as seen in the representative structures shown in *Figure 3a,d* and *Figure 3—figure supplement 1*. The fact that the dimensions of chains within the aggregate are much closer to the ideal state than the single chain in solution indicates that conformational disorder increases significantly upon aggregation. Together, these findings demonstrate how even aggregated peptide chains may approach a state of maximal conformational disorder.

## Peptide chain dynamics in solution and in the aggregate

The conformational ensembles of the peptide in solution and in the aggregate differ significantly with respect to chain dimensions (*Figures 1* and *5*), long-range contacts, hydrophobic interactions (*Figure 2*), and hydration (*Figure 3*). Despite these large global structural differences, the ensembles strongly resemble each other in terms of local secondary structure: the populations and lifetimes of the hydrogen-bonded turns are nearly identical for both SC and MC systems (*Table 1*). While the dynamics of turn formation are similar, non-local dynamics of the chain differs dramatically in solution and in the aggregate (*Video 1*). In particular, the lifetime of the open state (defined as the N- and C-termini not being in contact) increases more than fifty times upon aggregation (21 ± 1 ns vs. 1140 ± 30 ns in the SC and MC systems, respectively; *Figure 6*).

Reptation theory predicts characteristic signatures for the dynamics of polymer chains in melts (*de Gennes, 1979*): short timescale motions are predicted to obey Rouse-like dynamics, whereas long timescale motions should be strongly affected by the confinement imposed by neighbouring chains. To determine whether the dynamics of aggregated elastin peptides is characteristic of a melt, we analyzed the diffusion of the central residue of each chain (*Figure 6—figure supplement 1*). This residue exhibits anomalous diffusion (or sub-diffusion), with its mean-square displacement obeying a power law with exponent $\alpha = 0.58$. This value is similar to the exponent close to 0.6 found by Harmandaris et al. in atomistic simulations of polyethylene melts (*Harmandaris et al., 2003*) and is intermediate between 1/2 and 2/3, as expected respectively for a Rouse chain (*Rouse, 1953*; *Teraoka, 2002*) and for the Zimm model, an extension of the Rouse model that accounts for hydrodynamic interactions between chain monomers (*Teraoka, 2002*; *Zimm, 1956*). Importantly, we find no crossover to a regime with $\alpha = 1/4$, as one would expect if the chain were moving as if confined in a tube formed by the neighboring chains (that is, the entangled polymer melt regime described by de Gennes [*de Gennes, 1979*]). The chain length used here may be too short to observe this crossover.

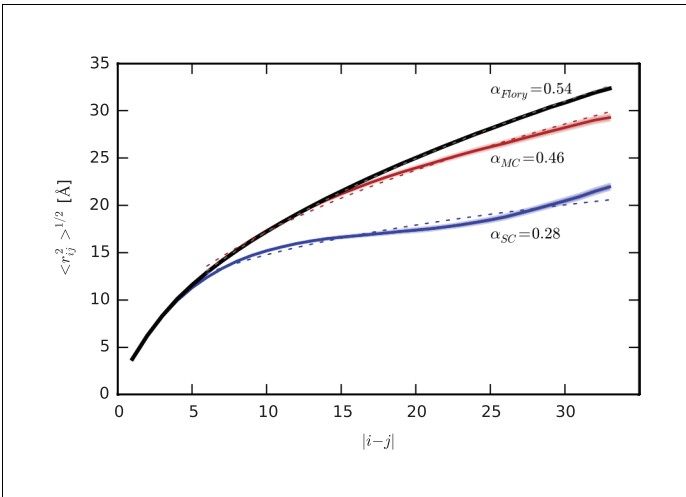

**Figure 5.** Intrachain distance scaling and comparison to the ideal, random-coil state. Root-mean-square distance, $<r_{ij}^2>^{1/2}$, between residues $i$ and $j$ as a function of sequence separation, $|i-j|$, for SC (blue) and MC (red), and for the ideal, random coil state modeled using the SC simulations according to the method of Flory *et al.* (black) (*Flory, 1969*; *Miller et al., 1967*) described in detail in **Supplementary file 1**. Shading indicates standard error. In each case, the dotted line indicates the power law fit to the data, with the exponent α provided next to each curve.

DOI: https://doi.org/10.7554/eLife.26526.015

The following figure supplements are available for figure 5:

**Figure supplement 1.** Equilibration of chain dimensions in the aggregate.
DOI: https://doi.org/10.7554/eLife.26526.016

**Figure supplement 2.** Temperature dependence of the conformational properties of the monomer.
DOI: https://doi.org/10.7554/eLife.26526.017

**Figure supplement 3.** Intrachain distance scaling for the ensemble obtained using TIP4P-D.
DOI: https://doi.org/10.7554/eLife.26526.018

---

Consistent with this hypothesis, Ramos *et al.* (*Ramos et al., 2016*) found no crossover for the shortest chain length of hydrogenated polybutadiene that they studied (36 monomers compared to 35 residues in the chain studied here). The absence of crossover may also be due to insufficient simulation length, or to the fact that the chains remain highly hydrated in the aggregate and are not characteristic of a solvent-excluding melt.

## Relevance to elastin-like peptides with cross-linking domains

Despite the moderate size of our aggregate, its melt-like properties suggest that the present study captures the fundamental basis for ELP phase separation. As such, the molecular basis for phase separation uncovered in this study is likely to be relevant to longer ELPs and full-length tropoelastin. In support of this point, our results are in excellent agreement with a recent NMR study of block peptides with alternating cross-linking domains and hydrophobic (GVPGV)₇ domains, successively in solution, in the coacervate, and in materials produced by cross-linking (*Reichheld et al., 2017*). The conformational ensemble of the hydrophobic domain presented in this study is consistent with the NMR results both qualitatively and quantitatively: (i) the peptides are disordered both before and after phase separation; (ii) the secondary structure in the hydrophobic domains consists primarily of sparse and transient β-turns in the VPGV and GVGV repeats, whose population was estimated to be in the range 20–40%, compared to our estimate of 16–20%; (iii) this secondary structure does not change significantly upon phase separation; (iv) phase separation entails formation of non-specific, intermolecular hydrophobic contacts; and (v) the protein-rich liquid phase is significantly hydrated. This broad agreement does not mean that the conformational ensembles of the hydrophobic domains are identical in the two model peptides (if only because of the different length of the polypeptide chains), but it indicates that the structural and physical basis for the self-assembly of the hydrophobic domains is not fundamentally affected by the presence of cross-linking domains.

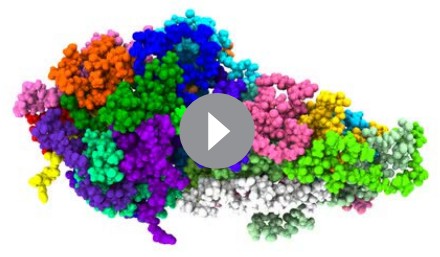

**Video 1.** In *Video 1*, the final 95 ns of a 5 microsecond trajectory of an aggregate is shown. Each of the 27 peptide chains is colored individually. Frames in the movie correspond to conformations separated by 50 ps time intervals. A smoothing window of 2 frames was applied and all conformations were aligned to the first conformation for clarity using VMD (*Humphrey et al., 1996*). As can be seen in the movie, both the global structure of the entire aggregate as well as the conformation of individual chains within the aggregate fluctuate on the nanosecond timescale. Chains on the surface of the aggregate occasionally extend outward into the surrounding water (water molecules are not shown for clarity).

DOI: https://doi.org/10.7554/eLife.26526.021

## Disordered aggregates: structure and function of self-assembled elastomeric proteins

The unusual properties of elastin and ELPs set them apart both from more common types of intrinsically disordered proteins (IDPs) that do not self-aggregate, and from proteins that form amyloid upon aggregation. On the one hand, the majority of IDPs have a high charge content and low sequence hydrophobicity, which allows them to avoid self-aggregation (*Uversky et al., 2000*), whereas elastin, because of its high content of hydrophobic residues, undergoes self-aggregation. Several computational studies have described structural ensembles of the more common type of IDPs. Among them, the RS peptide, which has a high net charge, was studied in detail using both simulations and experiments (*Rauscher et al., 2015*). Extensive computational studies of IDPs with varying fraction of charged residues have been carried out (*Das and Pappu, 2013*).

On the other hand, unfolded or misfolded proteins are prone to form ordered amyloid aggregates, but the amino acid composition of elastin precludes both folding and amyloid formation (*Rauscher et al., 2006*). While maintaining both disorder and hydration upon aggregation is crucial for elastic recoil, it is equally important that elastin and other self-assembled elastomeric proteins avoid the formation of the cross-β-structure characteristic of amyloid fibrils (*Rauscher et al., 2006*), which are postulated to be a thermodynamically stable state for any polypeptide chain under appropriate solution conditions (*Dobson, 2003*). Like the native state of globular proteins, the structure of amyloids is characterized by a water-excluding core and extensive backbone self-interactions. The liquid-like structure of elastin is incompatible with both protein folding and the formation of amyloid, and it is achieved through a high combined proportion of proline and glycine residues (*Rauscher et al., 2006*). Both proline, with its fixed φ dihedral angle and absence of amide hydrogen, and glycine, with its high entropic penalty for conformational confinement, inhibit the formation of α-helix and β-sheet structure, and serve to maintain a high degree of hydration and structural disorder by preventing the formation of a compact, water-excluding core. The essential role of proline and glycine in governing disorder, hydration, and elasticity was shown to extend to many other self-assembled elastomeric proteins (*Rauscher et al., 2006*), a finding corroborated by a recent study of an array of proline- and glycine-rich disordered proteins (*Quiroz and Chilkoti, 2015*) as well as by studies of various classes of spider silks (*Savage and Gosline, 2008a*; *Savage and Gosline, 2008b*). These results point to a fundamental relationship between sequence composition, conformational disorder, and elastomeric properties of self-assembled elastomeric proteins, including elastin, spider silk, and resilin.

Taken together, these considerations support a model of elastin aggregation and entropic elasticity as shown in *Figure 7*. This model highlights the contributions of the hydrophobic effect and conformational entropy to protein folding, aggregation, and elastic recoil. Although individual chains approach a state of maximal conformational disorder upon aggregation, excluded volume effects are expected to limit the overall conformational entropy of the aggregate because the conformations are dependent upon those of their neighbors in the protein-dense phase. As such, the present study cannot conclude on whether or not conformational entropy contributes to self-assembly. However, our results are qualitatively consistent with rubber-like elasticity, since the extension of disordered, cross-linked hydrophobic chains should lead to a decrease of conformational entropy of the

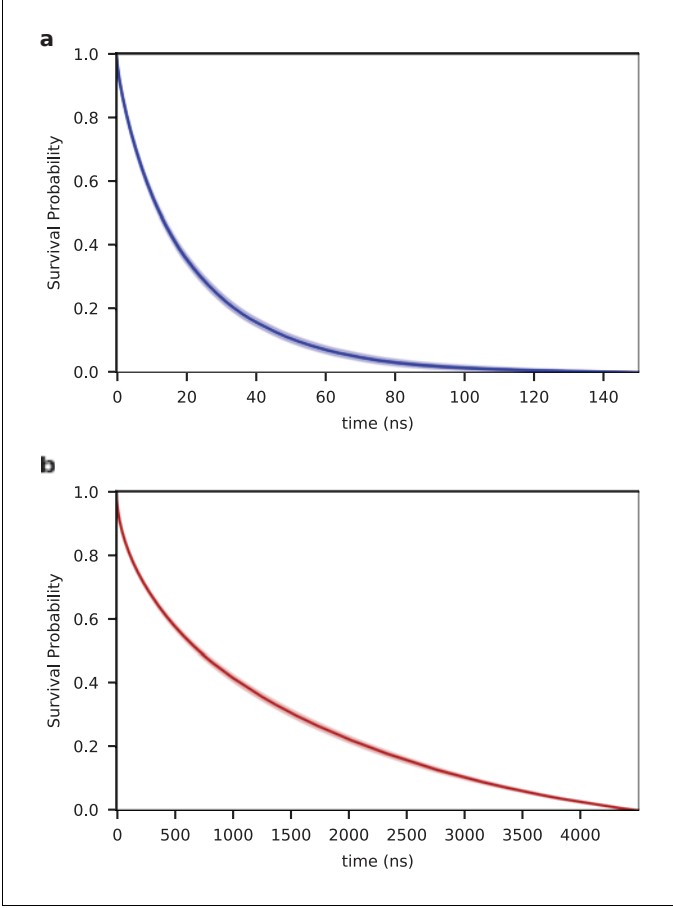

**Figure 6.** Kinetics of end-to-end contact formation. Survival probability of the open state (without a contact between the chain ends) as a function of time for the single chain (SC) (**a**) and for the aggregated chains (MC) (**b**) . The lifetime of the open state is $21 \pm 1$ ns and $1140 \pm 30$ ns for the SC and MC systems, respectively.
DOI: https://doi.org/10.7554/eLife.26526.019

The following figure supplement is available for figure 6:

**Figure supplement 1.** Chain dynamics within the aggregate.
DOI: https://doi.org/10.7554/eLife.26526.020

polymerized material as well as of the individual chains. Thus, both the hydrophobic effect and conformational entropy are expected to contribute to elastic recoil, reconciling the random coil and liquid drop models of elasticity.

## Conclusions

The present study provides the first atomistic description of a melt-like, disordered protein state, which may be called the liquid state of proteins. The model of elastin-like aggregates derived here represents the first detailed model of a protein coacervate. In spite of its moderate size, this molecular system emulates a biphasic liquid. Peptide aggregation is driven at least in part by the hydrophobic effect, which results in a three-fold increase in burial of non-polar groups for each polypeptide chain compared to the monomeric form. In the aggregate, the individual polypeptide chains approach a state of maximal conformational disorder as predicted by the Flory theorem. As such, the above results show how a classic concept of polymer theory, the polymer melt, is realized in an important but poorly-understood structural protein, and demonstrate the relevance of this concept to the self-assembly (coacervation) and mechanical properties of elastin.

Our results support a unified model of elastin structure and function that recapitulates experimental data and reconciles key aspects of previous qualitative models. The biological function of elastin

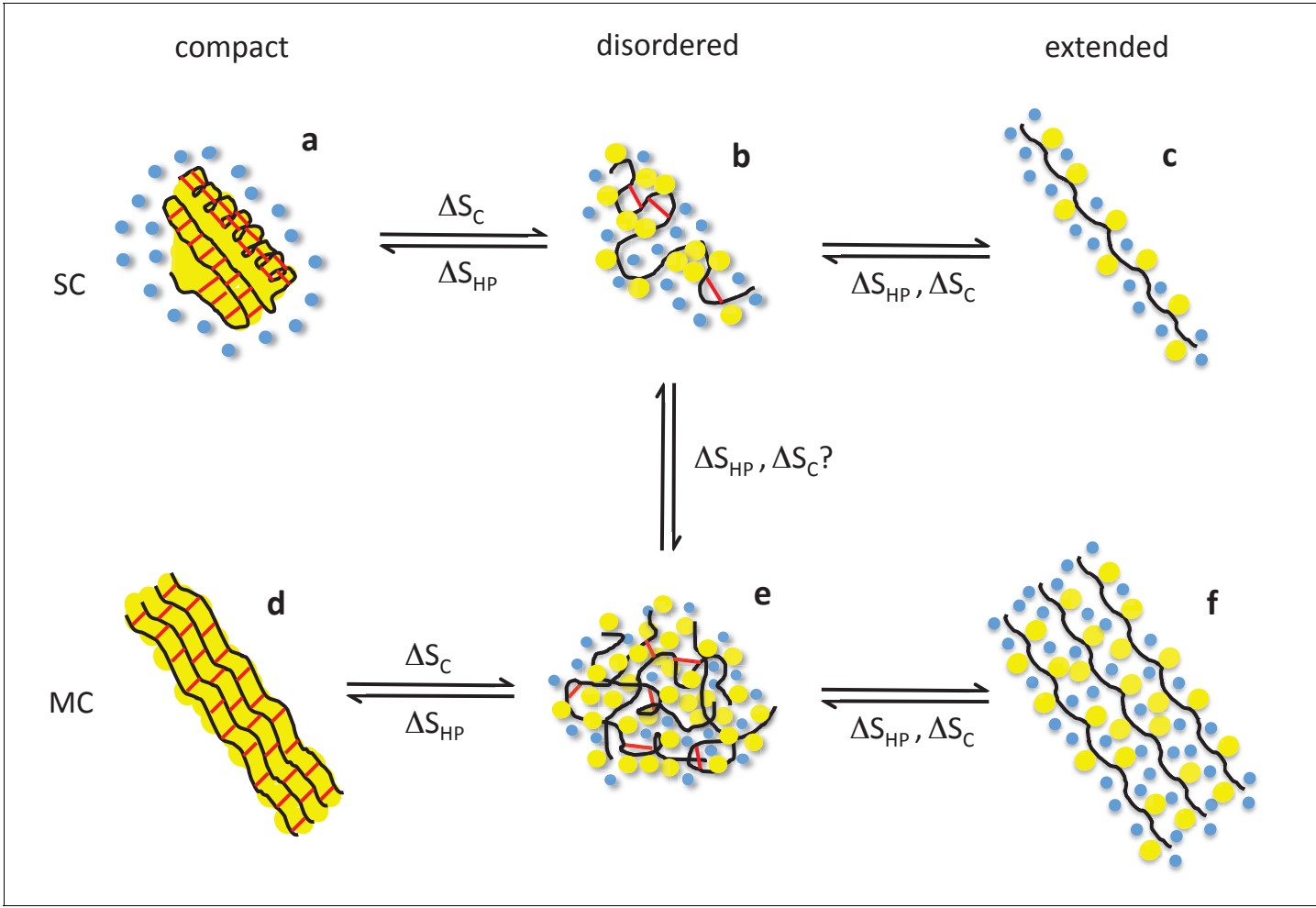

**Figure 7.** Structural basis of entropic elasticity in self-assembled elastomeric proteins. Schematic description of polypeptide main chains (black), non-polar side chains (yellow), solvating water molecules (blue), and peptide-peptide hydrogen bonds (red) in monomeric (SC, top row) and aggregated (MC, bottom row) states. Globular proteins that unfold or misfold are prone to aggregation, which leads to highly ordered amyloid fibrils. Both the native (**a**) and amyloid (**d**) states of globular proteins are characterized by extensive secondary structure and a water-excluding hydrophobic core. Despite their hydrophobic character, elastin and other self-assembled elastomers cannot form such compact structures due to their high content in proline and glycine. Instead, they are hydrated and disordered both in their monomeric (**b**) and aggregated (**e**) states, so that they may readily undergo extension and elastic recoil (**e-f**). The role of the two dominant types of entropy, the hydrophobic effect ($\Delta S_{HP}$) and chain entropy ($\Delta S_C$), is highlighted. While the hydrophobic effect favors hydrophobic collapse (c→b, f→e), aggregation (b→e), and, if possible, compact, water-excluding states (b→a, e→d), conformational entropy favors disordered (**b, e**) over extended (**c, f**) and compact (**a, d**) states. As a result, both entropic effects contribute to elastic recoil. Adapted with permission from *Rauscher et al. (2006)*, Structure (*Rauscher et al., 2006*).

DOI: https://doi.org/10.7554/eLife.26526.022

is incompatible with a unique, ordered structure. In the functional state, its hydrophobic domains form a water-swollen, disordered aggregate characterized by an ensemble of many degenerate conformations with significant backbone hydration and fluctuating local secondary structure. The combination of two entropic forces, the hydrophobic effect and polypeptide chain entropy, governs the elastic recoil central to elastin's function. These effects are intimately linked. Not only does the replacement of intramolecular contacts between non-polar side chains by intermolecular ones drive elastin self-assembly, it also helps the polypeptide chains of hydrophobic domains approach a state of maximal conformational disorder.

Remarkably, our findings defy conventional wisdom about protein folding, aggregation, and disorder: (1) although the structure of the aggregated peptide chains is nearly maximally disordered, it is not random but instead contains well-defined and significantly-populated secondary structure elements in the form of hydrogen-bonded turns; (2) however, because these turns are local, sparse, and

transient, the polypeptide backbone remains highly hydrated on average; as a result, (3) the hydrophobic side chains cannot form a compact, water-excluding core even though they are significantly buried.

The detailed model of the liquid phase of proteins obtained here for elastin is of direct relevance to the self-assembly and mechanical properties of other self-assembled elastomeric proteins, with which elastin shares a high content in proline and glycine (*Rauscher et al., 2006*; *Quiroz and Chilkoti, 2015*). In addition, by uncovering the structural and physico-chemical basis for the phase separation of elastin, this study also provides a frame of reference for understanding the phase separation of other functional disordered proteins, including the FG-nucleoporins that compose the selectivity barrier of the nuclear pore complex (*Patel et al., 2007*) and low-complexity protein assemblies implicated in the intracellular phase separation of membraneless organelles (*Toretsky and Wright, 2014*; *Nott et al., 2015*; *Elbaum-Garfinkle et al., 2015*; *Molliex et al., 2015*; *Feric et al., 2016*; *Brangwynne et al., 2015*). The basic mechanism of self-assembly uncovered in the present study, which entails replacement of weak, degenerate intramolecular interactions by intermolecular ones, is likely to apply to other disordered proteins undergoing single-component phase separation.

## Materials and methods

Atomistic MD simulations with explicit water were performed on an elastin-like peptide (ELP), (GVPGV)$_7$, successively as an isolated chain (single chain, SC) and an aggregate of twenty-seven chains (multi-chain, MC). This sequence, derived from a hydrophobic domain of chicken elastin, is the most extensively studied elastin repeat motif (*Muiznieks et al., 2010*). An accumulated simulation time of over 200 μs was required to reach statistical convergence. A description of simulation methods and structural analysis follows. We provide details of: MD simulations and analysis of the conformational ensembles, equilibration of the aggregate simulations, convergence of the simulations, and analysis of the interior/surface of the aggregate. A detailed description of the method used to model the random coil state is included as *Supplementary file 1*.

### Simulation details
#### Choice of force field
The CHARMM 22* force field (*Piana et al., 2011*) and the charmm-modified TIP3P model (*Jorgensen et al., 1983*; *MacKerell et al., 1998*) were used for the peptide and water, respectively. This combination of force fields was shown to produce conformational ensembles of an intrinsically disordered protein consistent with both SAXS and NMR measurements in an extensive force field comparison (*Rauscher et al., 2015*). This force field combination was also used in a landmark MD simulation study in which proteins of multiple structural classes were folded (*Lindorff-Larsen et al., 2011*). Recent studies of other intrinsically disordered proteins using the same force field also demonstrated a good agreement between simulation results and NMR data (*Stanley et al., 2014*; *Somavarapu and Kepp, 2015*).

In earlier MD simulation studies, we computed conformational ensembles of the (GVPGV)$_7$ peptide successively as a single chain (*Rauscher and Pomès, 2012*; *Rauscher et al., 2009*; *Rauscher et al., 2006*; *Rauscher and Pomès, 2010b*) and as an aggregate of eight chains (*Rauscher and Pomès, 2012*; *Rauscher, 2012*) using the OPLS-AA/L force field (*Kaminski et al., 2001*) together with the TIP3P water model (*Jorgensen et al., 1983*). Detailed structural properties of the peptide such as hydrogen bonding, non-polar contacts, and hydration propensities differ quantitatively between the ensembles obtained with different force fields; in particular, both SC and MC conformational ensembles obtained using OPLS-AA/L are significantly more collapsed than their counterparts obtained using CHARMM 22*. Nevertheless, it should be noted that all of the qualitative conclusions of the present work regarding the structural and physico-chemical basis of elastin self-aggregation are robust to the differences between these force fields.

To further examine the dependence of our results, particularly as regards the compactness of the isolated peptide in water, on the choice of force field, we repeated some of the simulations with a different force field combination. Specifically, we performed 18 repeats of a 0.5-μs MD trajectory using CHARMM22* with TIP4P-D, a modified water force field proposed by Piana *et al*. to improve the hydration of disordered protein ensembles (*Piana et al., 2015*). This new set of simulations leads

to a less-collapsed $(GVPGV)_7$ monomer (*Figure 1—figure supplement 2*) with size-scaling exponent 0.40, compared to the value of 0.28 obtained with the TIP3P water model (*Figure 5—figure supplement 3*). We note that the TIP4P-D force field was observed to destabilize the native state of two folded proteins (*Piana et al., 2015*). As such, TIP4P-D may underestimate hydrophobic interactions in the hydrophobic IDP studied here. Nevertheless, the fact that the scaling exponent of our monomeric ELP obtained with TIP4P-D is still less than the ideal limit is consistent with hydrophobic collapse and the overall conclusions of the paper. Consistent with the analysis of hydrogen bonding and non-polar contacts with charmm-modified TIP3P (*Figure 2*), GVGV and VPGV β-turns are still the most populated structures in the ensemble obtained using TIP4P-D (*Figure 2—figure supplement 4*). The main difference between the ensembles is a significant reduction in non-polar, non-local contacts in the TIP4P-D ensemble. We note that, despite quantitative differences in average structural properties, all of the qualitative conclusions of this work are consistent with the results of the additional simulations of the SC system carried out with CHARMM 22*/TIP4P-D.

## Simulation of the single chain (SC) system

The $(GVPGV)_7$ peptide was built in an extended conformation using the program UCSF Chimera (*Pettersen et al., 2004*). The simulation system consisted of the $(GVPGV)_7$ peptide with neutral termini (-NH2 at the N-terminus and -COOH at the C-terminus) in a rhombic dodecahedral box with 10815 water molecules and 0.15M NaCl. All simulations were carried out using GROMACS version 4.6 (*Hess et al., 2008*). Periodic boundary conditions were applied. The box was sufficiently large such that no contacts occurred between periodic images. Energy minimization was carried out using the steepest descent algorithm. The short-range electrostatic interactions and Lennard-Jones interactions were evaluated using a cutoff of 9.5 Å. Particle-mesh Ewald summation was used to calculate the long-range electrostatic interactions with a grid spacing of 1.2 Å and a fourth order interpolation (*Essmann et al., 1995*). The LINCS algorithm (*Hess et al., 1997*) was used to constrain covalent bonds and angles involving hydrogen atoms, and the SETTLE algorithm (*Miyamoto and Kollman, 1992*) was used to constrain bond lengths and angles of water molecules. Virtual sites were used (*Feenstra et al., 1999*), allowing the use of a 4 fs integration time step. The velocity rescaling thermostat was used for all MD simulations (*Bussi et al., 2007*). Equilibration simulations in the NPT ensemble were carried out for 10 ns using Berendsen pressure coupling (*Berendsen et al., 1984*) followed by 10 ns using the Parrinello-Rahman barostat (*Parrinello and Rahman, 1981*). The snapshot from this simulation with a volume closest to the average volume was then used for subsequent replica exchange (RE) (*Sugita and Okamoto, 1999*) simulations, which were carried out in the canonical ensemble. Sixty six temperatures between 298 K and 450 K were used. The chains are more expanded at higher temperatures (*Figure 5—figure supplement 2*), which leads to enhanced sampling. Each replica performed ~$0.5 \times 10^6$ attempted temperature jumps separated by 2 ps MD simulations. The total simulation time, including all temperatures, was 69.3 μs. Only the ensemble of conformations at 298 K was used for subsequent analysis. In addition to the replica exchange simulation, twenty 1 μs long simulations were carried out in the NPT ensemble at 298 K in order to carry out the analysis of chain dynamics and lifetimes of hydrogen-bonded turns. As described above, eighteen 0.5 μs simulations were carried out using the TIP4P-D water model (*Piana et al., 2015*) to further assess the dependence of the results on force field choice. All simulation parameters were kept the same, except that a larger simulation system with 16834 water molecules was used because the peptide is on average more expanded in the simulations with TIP4P-D than charmm-modified TIP3P.

## Simulation of the multi-chain (MC) aggregated system

The system of aggregated peptides was built by selecting 27 conformations at random from the single chain simulations. These conformations were placed on a 3×3×3 grid, maximally separated. The concentration of the system was 102.9 mg/mL, which is higher than the concentration needed to observe aggregation of similar elastin peptides (*Bellingham et al., 2001*). Thirty three replicate simulations were carried out. For each of these simulations, different random starting conformations were used. The simulation system in each case contained 27 peptides, ~39000 water molecules, and 0.15 M NaCl. All simulation methods were the same as for the single chain system, except that replica exchange was not used. Instead, long (5 μs) simulations were carried out in the NPT ensemble at

298 K using the Parrinello-Rahman barostat (following 5 ns of initial equilibration in the NVT ensemble). Including all replicate simulations, these production simulations were 165 µs in length.

## Simulation of water

In order to compare the water coordination distribution of the hydration shell to that of bulk water, we performed a canonical MD simulation of bulk water at 298 K. The simulation system consisted of ~14000 water molecules in a rhombic dodecahedral box. The charmm-modified TIP3P model (*Jorgensen et al., 1983*; *MacKerell et al., 1998*) for water was used. Following 5 ns of equilibration using Berendsen pressure coupling (*Berendsen et al., 1984*), production simulations were carried out for 200 ns using the Parrinello-Rahman barostat (*Parrinello and Rahman, 1981*). All simulation parameters were identical to the other simulations. Two water molecules are considered to be coordinated if their oxygen-oxygen distance is less than 3.5 Å (*Hummer et al., 1996*).

## Molecular visualizations, analysis and error estimation

The representations of conformations in *Figures 1a, b*, *3a and d*, and *Figure 3—figure supplement 1* were creating using Visual Molecular Dynamics (VMD) (*Humphrey et al., 1996*). Refer to *Supplementary file 1*, Table S2 for definitions of all interactions in *Figures 2* and *3*. All reported estimates of standard error for the SC system were obtained using a blocking procedure (*Flyvbjerg and Petersen, 1989*). All reported estimates of standard error for the MC system are standard errors of the mean, obtained by considering the 33 independent MD simulations as 33 independent measurements of the property of interest.

## Delineating the equilibration period of the aggregation simulations

The initial conformations for the MC simulations were selected at random from the replica exchange simulation of the single chain. These simulations begin with the 27 chains placed on a 3×3×3 grid. In the first part of the simulation, the chains associate to form an aggregate. The chains then rearrange within the aggregate to reach an equilibrium state. In order to delineate the equilibration phase of the simulation, we consider the running average of the radius of gyration and the total number of hydrogen bonds formed in the aggregate. This analysis of the collapse of the aggregates is provided in *Figure 1—figure supplement 1*. We find that the equilibration period represents a significant fraction of the simulation (50%, 2.5 µs). This long equilibration period is due to the slow process of the chains expanding within the aggregate, as well as finding optimal packing. While the chains readily form aggregates (nanosecond timescale), chain expansion (approaching the ideal, unperturbed state) occurs on the microsecond timescale in these simulations (*Figure 5—figure supplement 1*). Only the conformations accumulated after the equilibration period of each simulation are used for further structural analysis.

## Convergence of the aggregation simulations

The results reported in *Figures 1–6* are ensemble averages, which include the data from all 33 independent aggregation simulations. We also provide snapshots of the final conformation of 5 of the MC simulations (*Figure 3—figure supplement 1*), as well as hydrogen bonding and non-polar contact maps of each of the 33 replicates (*Figure 2—figure supplements 2* and *3*), which demonstrate that each of the individual simulations also exhibits similar structural properties compared to the average results reported in the main text. That is, each of the independent simulations also has similar conformational properties as the entire ensemble.

Long simulations of the aggregate were needed because of the slowed dynamics of interacting chains. To assess the degree to which chain dynamics were slowed upon aggregation, we analyzed the kinetics of end-to-end contact formation. The ends of the polypeptide chain are defined as being in contact if the C- and N- terminus are within a distance of 6 Å of each other. The lifetime of the open state was computed using the method outline by Yeh and Hummer (*Yeh and Hummer, 2002*). The survival probability for the SC and MC chains are shown in *Figure 6*. The lifetime of the open state is 21 ± 1 ns and 1140 ± 30 ns for the SC and MC systems, respectively.

## Analysis of the interior/surface of the aggregate

To assess whether or not there is a clear distinction between the interior and the surface of the aggregate, we computed the number of contacts between blob-sized segments of the chain and water molecules (*Figure 3—figure supplement 2b*). As a control, we carried out the same analysis for blob-sized segments in the single chain system. Consistent with Das and Pappu (*Das and Pappu, 2013*) and Pappu *et al* (*Pappu et al., 2008*)., we use a blob size of 5 residues (*Figure 3—figure supplement 2a*). This number corresponds to the spacing between proline residues in the ELP sequence. Note that the fits to the scaling profiles in *Figure 5* were carried out for $|i\text{-}j| > 5$ because the length of a blob-sized segment was found to be five residues, and the distance scaling within a blob differs from that outside of a blob.

## Acknowledgements

We thank FW Keeley, L Muiznieks, RV Pappu, and SG Whittington for discussions, and Q Huynh for help in the analysis. Computations were performed on the GPC supercomputer at the SciNet High-Performance Computing Consortium and on the supercomputer Guillimin from McGill University. SciNet is funded by the Canada Foundation for Innovation under the auspices of Compute Canada; the Government of Ontario; Ontario Research Fund - Research Excellence; and the University of Toronto. Guillimin is managed by Calcul Québec and Compute Canada; its operation is funded by the Canada Foundation for Innovation, NanoQuébec, RMGA and the Fonds de recherche du Québec - Nature et technologies. This work was supported by Canadian Institutes for Health Research operating grant MOP84496 to RP and by a Canada Graduate Scholarship from the Natural Sciences and Engineering Research Council and a Scholarship from the Research Training Centre at the Hospital for Sick Children to SR.

## Additional information

### Funding

| Funder | Grant reference number | Author |
|---|---|---|
| Canadian Institutes of Health Research | MOP84496 | Régis Pomès |
| Natural Sciences and Engineering Research Council of Canada | | Sarah Rauscher |
| Hospital for Sick Children | | Sarah Rauscher |

The funders had no role in study design, data collection and interpretation, or the decision to submit the work for publication.

### Author contributions

Sarah Rauscher, Conceptualization, Data curation, Software, Formal analysis, Validation, Investigation, Visualization, Methodology, Writing—original draft; Régis Pomès, Conceptualization, Resources, Supervision, Funding acquisition, Investigation, Methodology, Project administration, Writing—review and editing

### Author ORCIDs

Sarah Rauscher [iD] http://orcid.org/0000-0001-9860-3237
Régis Pomès [iD] http://orcid.org/0000-0003-3068-9833

### Decision letter and Author response

Decision letter https://doi.org/10.7554/eLife.26526.027
Author response https://doi.org/10.7554/eLife.26526.028

# Additional files

## Supplementary files

• Supplementary file 1. Supplementary information: Supplementary Methods (Method to Compute the Average Dimensions of the Ideal, Random Coil State), Figure S1 and Tables S1 and S2.
DOI: https://doi.org/10.7554/eLife.26526.023

• Supplementary file 2. The supplementary file contains a coordinate file (prot_100ns.gro) and a trajectory file corresponding to the final 100 ns (4_9 ms_to_5 ms.xtc) of one of the aggregate simulations. A larger trajectory file (4 ms_to_5 ms.xtc) corresponding to the final microsecond of the same simulation can be downloaded from the public repository figshare.com (S. Rauscher and R. Pomès, 'Microsecond-long molecular dynamics trajectory of an aggregate of elastin-like peptides in water.' DOI: 10.6084/m9.figshare.5532214).
DOI: https://doi.org/10.7554/eLife.26526.024

## Major datasets

The following dataset was generated:

| Author(s) | Year | Dataset title | Dataset URL | Database, license, and accessibility information |
|---|---|---|---|---|
| S Rauscher, R Pomès | 2017 | Microsecond-long molecular dynamics trajectory of an aggregate of elastin-like peptides in water | https://dx.doi.org/10.6084/m9.figshare.5532214.v1 | Available at figshare under a CC0 Public Domain licence (https://figshare.com/). |

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
