## [Decision Letter]

Thank you for submitting your article "The Liquid Structure of Elastin" for consideration by *eLife*. Your article has been reviewed by three peer reviewers, and the evaluation has been overseen by a Reviewing Editor and Arup Chakraborty as the Senior Editor. The following individual involved in review of your submission has agreed to reveal his identity: Robert Best (Reviewer #1).

The reviewers have discussed the reviews with one another and the Reviewing Editor has drafted this decision to help you prepare a revised submission.

Summary:

The paper uses molecular simulations to study the properties of elastin. In extensive simulations of elastin fragments with a simple repeat sequence, the authors find that the peptides are disordered and self-associate to form a dense protein-rich phase. This appears to be the first study that uses all-atom simulations to tackle the formation of liquid-like protein aggregates. The results support many of the features of elastin determined experimentally such as formation of local turn structure, but do not support an overall ordered aggregate which had been proposed in the past. Instead, individual peptides are still able to diffuse, albeit slowly.

In their reports, the three reviewers raise a number of serious concerns. The main criticisms are that (79) the simulation systems are highly simplified and may not adequately capture all relevant properties of real elastin, (12) the significance of the results is not clear because important negative controls and reference simulations are missing, (3) possible force field issues are not assessed, and (71) the comparison to experiment is weak. A detailed list of the criticisms follows below. These points would have to be addressed in a revision.

Essential revisions:

1) The simulation systems are highly simplified and may not adequately capture all relevant properties of real elastin. Peptides with a simple repeat sequence (VPGVG)_7_ are studied. Whereas the pentapeptide and related polypentapeptides have been accepted as suitable mimics for the extensive hydrophobic regions of elastin, it is important to discuss the advantages and possible drawbacks and limitations stemming from the choice of sequence. Of particular concern is the fact that the native protein is largely characterized by alternating hydrophobic and crosslinking domains. Although the hydrophobic domains have sequences similar to the model of this study, they are usually flanked by crosslinking domains that have at least partial α-helical character. As such, it would appear rather unlikely that 27 hydrophobic domains in the native protein would form an aggregate like the one used here.

2) The significance of the results is not clear because important negative controls are missing. Wouldn't any concentrated solution of peptides with a roughly similar sequence composition behave in the same way, i.e., form a solvated blob? It would help to contrast the elastin results to simulations of IDP peptide fragments that do not aggregate and of IDPs that form "solid" aggregates in simulations using a similar protocol. In particular, is the extent of hydration in the core of the elastin bundle significantly higher than for other aggregating peptides? The authors mention silk and amyloid as two extremes of hydrated and solid aggregates. There are even reports in the literature that show a tendency to fibrillize (Fred W. Keeley, Catherine M. Bellingham and Kimberley A. Woodhouse, Philosophical Transactions: Biological Sciences, Vol. 357, No. 1418, Elastomeric Proteins: Structures, Biomechanical Properties and Biological Roles (Feb. 28, 2002), pp. 185- 189).

3) The choice of force field seems problematic, because it has been reported to favor overly compact structures in other systems. See e.g. Figure 1 in Piana et al., Curr. Opin. Struct. Biol. v24, p98, 2014, where the scaling of Rg with number of residues comes close to the one-third power, whereas most protein chains in water are closer to ideal chain scaling. Now elastin is quite hydrophobic, so it is possible that all is fine and it should be collapsed anyway, but it would be good to have some discussion of this issue and maybe a comparison with experiments which show that the elastin monomer behavior is reasonable.

4) The comparison to experiment is rather weak and qualitative at best, and therefore should be strengthened. For instance, can the simulations be related quantitatively to the neutron scattering experiments in Perticaroli et al., 2015? In light of the considerable force field issues in IDP simulations (well documented by work also of the authors!), it is very important to have experimental support. Comparisons to specific experiments should be performed in a way that could, at least in principle rule out, this and/or other models.

---

## [Author Response]

Essential revisions:1) The simulation systems are highly simplified and may not adequately capture all relevant properties of real elastin. Peptides with a simple repeat sequence (VPGVG)_7_ are studied. Whereas the pentapeptide and related polypentapeptides have been accepted as suitable mimics for the extensive hydrophobic regions of elastin, it is important to discuss the advantages and possible drawbacks and limitations stemming from the choice of sequence. Of particular concern is the fact that the native protein is largely characterized by alternating hydrophobic and crosslinking domains. Although the hydrophobic domains have sequences similar to the model of this study, they are usually flanked by crosslinking domains that have at least partial α-helical character. As such, it would appear rather unlikely that 27 hydrophobic domains in the native protein would form an aggregate like the one used here.

Indeed, the present model system is not designed to capture all the properties of elastin. Instead, we examine the structural and physical basis for the phase separation of hydrophobic elastin-like domains, which also provides insight into the structural basis of elasticity. Although cross-linking domains are required to form elastomeric materials, to which they confer integrity and durability, they do not undergo phase separation on their own and are not required for phase separation. Numerous studies from the labs of Tony Weiss, Fred Keeley, and Tony Tamburro over the past couple of decades have established that hydrophobic domains drive the phase separation of full-length tropoelastin. Accordingly, elastin-like peptides modeled on hydrophobic domains of tropoelastin but lacking cross-linking domains coacervate on their own. In addition, the hydrophobic domains are also thought to confer elastic recoil to elastin. Together with computational feasibility, these are the reasons why we chose to model hydrophobic domains. **–** See revised text in the first paragraph of the Introduction.

Inasmuch as hydrophobic domains drive the phase separation of full-length tropoelastin and the elasticity of polymerized elastin, it is likely that the molecular basis for phase separation and elastic recoil uncovered in this study is relevant to full-length tropoelastin. This point is corroborated by a recent NMR study of the self-assembly and polymerization of elastin-like block-copolymer peptides consisting of alternating cross-linking and (GVPGV)_7_ domains (Reichheld et al., PNAS May 2017). The structure of our isolated (GVPGV)_7_ domains is essentially preserved in the block copolymers: they are disordered, hydrated, and retain the same local secondary structure, which consists of transient β-turns, in soluble and phase-separated forms; and they form extensive, non-specific hydrophobic contacts upon phase separation.

This broad agreement does not mean that the conformational ensembles of the hydrophobic domains are identical in the two model peptides (if only because of the different length of the peptides), but it indicates that the essential properties of the hydrophobic domains are not fundamentally affected by the presence of cross-linking domains. Moreover, despite the small size of our aggregate, its liquid or melt-like behavior suggests that the present study captures the fundamental basis for ELP phase separation. See the new Results section titled “Relevance to elastin-like peptides with cross-linking domains” and our response to point 4 below.

2) The significance of the results is not clear because important negative controls are missing. Wouldn't any concentrated solution of peptides with a roughly similar sequence composition behave in the same way, i.e., form a solvated blob? It would help to contrast the elastin results to simulations of IDP peptide fragments that do not aggregate and of IDPs that form "solid" aggregates in simulations using a similar protocol. In particular, is the extent of hydration in the core of the elastin bundle significantly higher than for other aggregating peptides? The authors mention silk and amyloid as two extremes of hydrated and solid aggregates. There are even reports in the literature that show a tendency to fibrillize (Fred W. Keeley, Catherine M. Bellingham and Kimberley A. Woodhouse, Philosophical Transactions: Biological Sciences, Vol. 357, No. 1418, Elastomeric Proteins: Structures, Biomechanical Properties and Biological Roles (Feb. 28, 2002), pp. 185- 189).

Previous studies provide compelling evidence for the special properties of elastin and other self-assembled elastomeric proteins as opposed to both (a) more common types of IDPs that do not self-aggregate and (b) proteins that form amyloid-like aggregates:

a) The majority of IDPs have a high charge content and low sequence hydrophobicity, which allows them to avoid self-aggregation (Uversky, 2000), whereas elastin, because of its high content of hydrophobic residues, undergoes self-aggregation. Several computational studies have described structural ensembles of the more common type of IDPs. Among them, the RS peptide, which has a high net charge, was studied in detail using both simulations and experiments (Rauscher et al., JCTC 2015). Extensive computational studies of IDPs with varying fraction of charged residues have been carried out (Das and Pappu, 2013).

b) Unfolded or misfolded proteins are prone to form ordered amyloid aggregates, but, as we showed a decade ago, the amino acid composition of elastin is designed to avoid the amyloid fate (Rauscher et al., JCTC 2006). In that joint computational and experimental study with the Keeley lab, we showed that contrary to amyloidogenic peptides, self-assembled elastomeric proteins, including elastin and spider silks, remain hydrated and disordered even in the aggregated state. These properties are manifested in an apparent threshold in combined glycine and proline composition above which amyloid formation is impeded and elastomeric properties become apparent. High P and G content prevents the polypeptide backbone of self-assembled elastomeric proteins from adopting regular secondary structure required to form the water-excluding hydrophobic cores (in which the core is essentially completely dehydrated) found in globular proteins and in the core structure of amyloid fibrils—ensuring instead that the backbone is hydrated and disordered.

To clarify these points and provide more context on elastin as an aggregating/phase separating IDP, we have added a new section titled “Disordered aggregates: Structure and function of self-assembled elastomeric proteins” to the Results and Discussion, together with a new figure emphasizing the role of the hydrophobic effect and conformational disorder in protein folding, assembly, and elastic recoil (Figure 7). In addition, we added analysis of the density profile of peptide and water in our system (subsection “Hydration and disorder of the polypeptide backbone”, last paragraph, Figure 4), which shows that the peptides retain over 20% internal hydration in both monomeric and aggregated states.

3) The choice of force field seems problematic, because it has been reported to favor overly compact structures in other systems. See e.g. Figure 1 in Piana et al., Curr. Opin. Struct. Biol. v24, p98, 2014, where the scaling of Rg with number of residues comes close to the one-third power, whereas most protein chains in water are closer to ideal chain scaling. Now elastin is quite hydrophobic, so it is possible that all is fine and it should be collapsed anyway, but it would be good to have some discussion of this issue and maybe a comparison with experiments which show that the elastin monomer behavior is reasonable.

We have modified the text to address and clarify the force field issue as explained below:

a) We cannot compare our results to experimental measurements performed on the same peptide, because to our knowledge such data are lacking for the (GVPGV)_7_ monomer. However, our results are supported by NMR measurements performed on an elastin-like peptide containing hydrophobic domains with the same sequence. See our response to points (79) and (71) and revised text in the Results section titled “Relevance to elastin-like peptides with cross-linking domains”.

b) Our choice of the CHARMM22* and CHARMM-refined TIP3P force fields is directly motivated by Rauscher et al. (JCTC 2015), which shows that out of 8 combinations of protein and water force fields tested, this particular combination best reproduced experimental data on an intrinsically disordered peptide. See the first paragraph of the Materials and methods section titled “Choice of Force Field”.

c) We initially performed this study several years ago using a different force field combination (OPLS/AA with TIP3P) (Sarah Rauscher’s PhD thesis, 2012). This previous study led to significantly more compact estimates of the radius of gyration of the peptide monomer. As is now well appreciated, OPLS tends to overestimate the degree of collapse or compaction of unfolded or disordered states of proteins. Nevertheless, these results were qualitatively similar to those of the present study: disordered and hydrated peptides characterized by transient β-turns approached a melt-like state upon self-assembly. See the revised text in the last paragraph of the Materials and methods section titled “Choice of Force Field”.

d) In addition, to fully address the request to examine the dependence of our results, particularly as regards the size of the peptide monomer in water, on the choice of force field, we have repeated some of the simulations with a different force field combination. Specifically, we have performed 18 repeats of a 0.5-µs MD trajectory using CHARMM22* with TIP4P-D, a modified water force field with increased solvent-solute dispersion interactions recently proposed by Piana et al. (J. Phys. Chem. B, 2015) to improve the hydration of disordered protein ensembles. This new set of simulations leads to a less-collapsed (GVPGV)_7_ monomer with size-scaling exponent 0.40, compared to the value of 0.28 obtained with the TIP3P model. We note that the TIP4P-D force field was observed to destabilize the native state of two folded proteins (Piana et al., 2015). As such, TIP4P-D may underestimate hydrophobic interactions in the hydrophobic IDP studied here. Nevertheless, the fact that the scaling exponent of our monomeric ELP obtained with TIP4P-D is still less than the ideal limit is consistent with hydrophobic collapse and the overall conclusions of the paper. The analysis of these new simulations is shown in Figure 1—figure supplement 2, Figure 2—figure supplement 4, and Figure 5—figure supplement 3, and further discussion of force field dependence was added to the Materials and methods section (“Choice of force field”, last paragraph).

Together, points (c) and (d) indicate that our conclusions regarding the structural and physical basis for phase separation of hydrophobic elastin-like peptides are robust to the choice of force field, whether or not the compactness of the aqueous monomer is over- or underestimated.

4) The comparison to experiment is rather weak and qualitative at best, and therefore should be strengthened. For instance, can the simulations be related quantitatively to the neutron scattering experiments in Perticaroli et al., 2015? In light of the considerable force field issues in IDP simulations (well documented by work also of the authors!), it is very important to have experimental support. Comparisons to specific experiments should be performed in a way that could, at least in principle, rule out this and/or other models.

As mentioned above, experimental measurements for (GVPGV)_7_ monomers are lacking. However, the results presented in this manuscript are in excellent agreement with an NMR study by Reichheld et al. (2017), which was published after this manuscript was submitted. The NMR study describes the structure and dynamics of block peptides with alternating cross-linking domains and hydrophobic domains with the same sequence as in the present study, successively in solution, in the coacervate, and in materials produced by cross-linking. The conformational ensemble of the hydrophobic domain presented in this study is consistent with the NMR results, both qualitatively and quantitatively. In particular, (i) the peptides are disordered both before and after phase separation; (ii) the secondary structure in the hydrophobic domains consists primarily of sparse and transient β turns in the VPGV and GVGV repeats, whose population was estimated to be in the range 20-40%, compared to our estimate of 16-20%; (iii) this secondary structure does not change significantly upon phase separation; (iv) phase separation entails formation of non-specific, intermolecular hydrophobic contacts; and (v) the protein-rich liquid phase is significantly hydrated.

This agreement corroborates the present results and indicates that the structural and physical basis for the self-assembly of hydrophobic domains is not fundamentally affected by the presence of cross-linking domains. We have added this experimental corroboration to the revised text (section “Relevance to elastin-like peptides with cross-linking domains”).